# To Volunteer or Not? Perspectives towards Pre-Registered Nursing Students Volunteering Frontline during COVID-19 Pandemic to Ease Healthcare Workforce: A Qualitative Study

**DOI:** 10.3390/ijerph18126668

**Published:** 2021-06-21

**Authors:** Betsy Seah, Ben Ho, Sok Ying Liaw, Emily Neo Kim Ang, Siew Tiang Lau

**Affiliations:** Alice Lee Centre for Nursing Studies, Yong Loo Lin School of Medicine, National University of Singapore, Clinical Research Centre, Block MD11, Level 2, 10 Medical Drive, Singapore 117597, Singapore; benho@u.nus.edu (B.H.); nurliaw@nus.edu.sg (S.Y.L.); nuranke@nus.edu.sg (E.N.K.A.); nurlst@nus.edu.sg (S.T.L.)

**Keywords:** COVID-19, health workforce, nursing students, professional identity, qualitative study, volunteers

## Abstract

COVID-19 has caused a shortage of healthcare workers and has strained healthcare systems globally. Pre-registered healthcare students with training have a duty of care and can support the healthcare workforce. This study explored factors influencing the willingness of final-year nursing students to volunteer during the COVID-19 pandemic, the role of professional identity in volunteering as healthcare workers, and strategies to improve future volunteering uptakes and processes. A qualitative study using focus-group discussions was conducted. Final-year nursing students who volunteered, students who did not volunteer, and lecturers who supervised student volunteers were recruited. Interviews were conducted online, video-recorded, and transcribed verbatim. A thematic analysis was used. The themes were “wavering thoughts on volunteering”, “bringing out ‘the nurse’ in students through volunteering” and “gearing up to volunteer”. Findings suggested the need to look beyond the simplicity of altruism to the role of professional identity, operational, and motivational factors to explain nursing students’ decision to volunteer and their volunteer behavior. Providing accommodation, monetary and academic-related incentives, supporting the transitionary phase from students to “professional volunteers”, promoting cohesive and positive staff–student volunteer relationships, and establishing a volunteer management team are strategies identified to improve volunteering uptake and operational processes. Our findings advocate strategic partnerships between hospitals/communities and academic institutions in providing various healthcare services during pandemics.

## 1. Introduction

The 2019 novel coronavirus (COVID-19) pandemic has become a major health crisis affecting millions of people across the globe [1]. To manage the surge of COVID-19 patients, there was a sudden shortage of healthcare workers, and this strained healthcare systems globally. Additional roles, such as screening for potential cases, implementing quarantine and contact tracing contributed further pressures on the healthcare workforce.

In times of crisis, healthcare professionals, government agencies and volunteers from official agencies would form the formal emergency and disaster management system. However, the willingness of healthcare workers to work during a respiratory disease pandemic ranged between 23.1% and 95.8%, and was influenced by various factors such as gender, profession, perceived personal safety, awareness of pandemic risk, role-specific knowledge, pandemic response training and confidence in personal skills [2]. On the other hand, individuals volunteering their time, knowledge, skills, and resources will obtain legitimacy and become part of the formal system through agencies’ volunteer recruitment [3]. Factors influencing the willingness of healthcare professionals to serve on the front line have an impact on manpower supply. Older ages, psychological stress, previous volunteering experience, perceived knowledge of disease prevention, safety concerns, and social support were associated with a willingness to serve on the front line [4,5,6].

Due to the healthcare workforce shortage, healthcare students are perceived to have a moral and professional obligation to volunteer, and are strongly encouraged to do so during pandemics [5]. Pre-registered healthcare students, when equipped with the right skills and knowledge, can render great support to the formal healthcare workforce. During the COVID-19 pandemic, medical students in Demark and the United Kingdom were mobilized as temporary residents, ventilator therapy assistants, or nursing assistants [7]. Similarly, final-year nursing students in Spain were deployed to work in hospitals to reinforce staffing levels, often remunerated similarly to nursing care assistants [8]. These students felt highly committed to volunteer and worked as nurses despite the uncertainty and non-completion of their nursing course [8]. However, the willingness of students to join the workforce was dependent on factors such as a sense of duty, perceived risk of infection, personal health, lack of protocol and knowledge, and perceived preparedness and skill competency required to execute volunteer duties [8,9,10].

In Singapore, the COVID-19 crisis evolved from multiple community cluster outbreaks in February 2020 to large endemic spreads in migrant worker dormitories, beginning in April 2020. These dormitories were high-density communal residences, with unhygienic living conditions for low-waged migrant workers who lacked equal access to healthcare and social safety-nets, and these factors elevated the risk of large outbreaks of respiratory diseases [11]. Hundreds of new migrant worker cases were detected daily, with numbers soaring to a peak of 1397 on 20 April 2020 [12]. A national task force was established to coordinate Singapore’s outbreak response. As part of the public health strategy, mass testing facilitated the quick identification of cases, immediate isolation of cases, contact tracing and precautionary self-isolation of close contacts. This required trained personnel to conduct nasal swabs and serology tests on both infected and non-infected individuals, and placed manpower strains on Singapore’s healthcare system. Apart from hiring short-term nasopharyngeal swabbers and swabbing assistants, Singapore’s Ministry of Health sought volunteers from current and former healthcare professionals, as well as healthcare students, to support these testing efforts.

Eighty-five pre-registered final-year nursing students from a university in Singapore were approached in May 2020 to volunteer and perform venipuncture for migrant workers. Among them, only 27 (31%) volunteered. As competency in venipuncture was not part of the pre-registration undergraduate nursing curriculum in the university, these students underwent a phlebotomy course conducted by the university’s nursing faculty, where they practiced venipuncture using simulated intravenous arms. Personal protective equipment (PPE) training and mask refitting were also conducted. These students were then deployed to the dormitories and received on-the-job training, with close supervision by a team of doctors, nurses, and phlebotomists from a hospital cluster. The student volunteers had to perform 20 successful venipunctures to certify their competency in phlebotomy. Thereafter, they would volunteer their phlebotomy services for six weeks. Additionally, three nursing faculty members, who also volunteered to be frontline phlebotomists, supervised or worked alongside these student volunteers.

Although the uptake of volunteering was similarly reported in Saudi Arabia [9], the authors expected the numbers to be higher, and wanted to understand how to improve future volunteering uptake and processes. By understanding the factors influencing pre-registered healthcare students’ willingness to volunteer and their volunteer experience, healthcare leaders and volunteer managers can better facilitate manpower planning and volunteer recruitment for future crisis management. From an educational perspective, learning about students’ decision-making on volunteering and their volunteering experiences in relation to professional identity may help to shape curriculum planning and enhance learning experiences. This study was thus undertaken to (a) explore factors influencing final-year nursing students’ willingness to volunteer during the COVID-19 pandemic, (b) explore the role of professional identity in volunteering as a frontline healthcare worker among final-year nursing students, and (c) identify strategies to improve future volunteering uptakes and processes.

## 2. Methods

### 2.1. Design and Procedure

We performed a qualitative study and collected data via focus-group discussions (FGDs). Using a semi-structured interview guide, seven FGDs were conducted online via Microsoft Teams at the nursing institute of a Singapore university between September 2020 and November 2020. Prior to the start of data collection, the study protocol was approved by the University’s Institutional Review Board ethics committee (IRB-2020-176). Invitation emails to participate in this study were sent to all fourth-year undergraduate nursing students, and to three lecturers who provided supervision to the student volunteers. Recruitment calls were also broadcasted by the researchers during the students’ e-lecture. Interested participants contacted the research team and provided their personal contact details to arrange for the conduct of FGDs. Written informed consent was obtained from all study participants.

### 2.2. Participants

A total of 33 participants were recruited via purposive sampling and were grouped as follows: students who volunteered as frontline phlebotomists (3 FGDs, *n* = 15), students who did not volunteer (3 FGDs, *n* = 15), and lecturers who provided supervision to the student volunteers (1 FGD, *n* = 3). All 85 final-year undergraduate nursing students who were invited to volunteer as frontline phlebotomists between June and July 2020 were eligible to participate in this study. The three nursing faculty members who supervised or worked alongside the student volunteers also participated in this study. Individuals who refused to be video-recorded on Microsoft Teams were excluded. The demographic characteristics of the participants are presented in Table 1. None of the participants dropped out of the study.

### 2.3. Data Collection

The research team developed an interview guide to explore the following topics: (a) the willingness of final-year nursing students to volunteer as frontline healthcare workers during the COVID-19 pandemic, (b) nursing students’ consideration factors for volunteering or not volunteering as frontline healthcare workers, (c) the role of professional identity in influencing the decision and experience of volunteering as frontline healthcare workers, and (d) areas of improvement to increase voluntary participation as frontline healthcare workers and to enhance the volunteering experience. A pilot FGD involving five nursing students was conducted to test the questions and determine if they addressed the identified research aims. Data shared during the pilot FGD was not used for the analysis of this study. As the concept of professional identity was found to be abstract during the pilot FGD, several questions were rephrased to contextualize or personalize the act of volunteering, to elicit responses that addressed the study’s aims. Examples of revised questions included: “Do you think nursing students are professionally obligated to volunteer in times of pandemic, and why?”, and “Does your volunteering experience change the way you view yourself as a nurse?”. The interview guide is presented in Table 2.

All FGDs were moderated by the same researcher (B.S.); B.S. was accompanied by either colleague S.T.L. or S.Y.L., who co-moderated the sessions, took down field notes during the FGDs, and summarized content shared by participants at the end of each FGD. All the moderators (B.S., S.T.L. and S.Y.L.) were females, had completed a Ph.D., and had experience in conducting qualitative research. They were known to the participants as faculty members of the undergraduate nursing program. Where existing dependent and/or explicit relationships were involved, the researcher would not be involved in the FGD. No other persons were present except for the participants and the identified researchers. All interviews were video-recorded. The average duration of each online FGD was 92 min. No repeat focus-group interviews were conducted. Prior to the FGDs, participants completed a brief socio-demographic questionnaire. Memos were written by B.S. after the conduct of each FGD; they were shared with the two co-moderators and were used to assess data saturation.

### 2.4. Data Analysis

All the FGDs were transcribed verbatim. None of the transcripts were returned to participants for checking. Qualitative data were analyzed inductively using thematic analysis [13]. Coding was conducted independently by two researchers, B.S. and B.H. Two coders (B.S. and B.H.) first familiarized themselves with the data by reading and re-reading all transcripts and listened to all interview recordings. This was followed by independent coding of all transcripts with the aid of N-Vivo version 12 software and a Microsoft Excel spreadsheet, by B.S. and B.H., respectively. The two coders discussed their initial semantic codes with each other to resolve differences before they reduced these initial codes to latent codes, and to preliminary sub-themes and themes together. The entire process of data reduction was iterative. Subsequently, a consensus meeting with S.T.L. was held to reach an agreement on data interpretation and refine the sub-themes and themes.

## 3. Results

We identified three themes. They include (1) wavering thoughts on volunteering, (2) bringing out “the nurse” in students through volunteering, and (3) gearing up to volunteer. Table 3 depicts an overview of themes, sub-themes, and reduced codes.

### 3.1. Wavering Thoughts on Volunteering

The first theme revealed the wavering thought processes of final-year nursing students making the decision to volunteer as frontline phlebotomists during the COVID-19 pandemic. Student participants found themselves in a dilemma, weighing their intrinsic motivations and the extrinsic concerns of volunteering. Although many were intrinsically motivated to volunteer and help, they wanted to *“know what I’m getting into before I make a responsible accountable decision”* as they had to consider *“the stakeholders involved and whether we would actually be helping when we offer our services to volunteer”*. Only a few of the student volunteers were not constrained by environmental considerations and stepped forward without hesitation.


*“(It) was a very, very difficult decision… like how difficult I would say is the email has a deadline for us to submit, right? I submitted like only 10 min before the deadline because I was deciding right up till that point.”*
(FGD1, P2, volunteer)

#### 3.1.1. Propelling Intrinsic Motivators

Participants expressed that they volunteered because they wanted to contribute and help others; some were motivated by altruism, interest, and passion. A few mentioned that it was a calling, and they felt a responsibility to step up as future healthcare personnel. Three participants mentioned that they were looking for volunteering opportunities, even before the school approached them. Some participants were concerned about the mental health and welfare of migrant workers and wanted to help them. Two lecturers commented that personality played a great role; they observed that students who volunteered were more outspoken, outgoing, adventurous, and willing to learn.

As venipuncture was considered a new clinical skill, student participants had to appraise their confidence and skill competence. They recognized that confidence and clinical competence varied across peers. Some were afraid their inexperience and inabilities would burden the strained healthcare workforce. Some participants had the courage to volunteer, while others did not.


*“There’s a lot of expectations for us to be competent… the nurses, physicians and all…they are already very stressed, so we don’t want to burden the rest also...”*
(FGD1, P1, volunteer)


*“I feel uncertain… because… we just completed our (clinical) attachment (placement) and personally, I feel I’m not competent enough to enter a pandemic situation when my competency in a clinical setting isn’t stable.”*
(FGD6, P1, non-volunteer)

Student volunteers identified volunteering as a great opportunity to apply nursing knowledge and skills in *“real-life scenarios”*. They looked forward to seeking new, interesting, and meaningful learning experiences. A few mentioned that the opportunity to experience *“first-hand on the ground” “how a hot zone area works”* during pandemics was *“once in a lifetime”*. While some non-volunteers wanted a break from their clinical placement, others viewed volunteering as a purposeful activity to pass the time and avoid boredom at home. A few participants commented that some of their peers did not volunteer because they were uninterested in pursuing a career in nursing upon graduation.

#### 3.1.2. Accounting for Extrinsic Concerns

Participants commented that the lack of information on disease transmission and management during the early phases of the pandemic brought about uncertainty, anxiety, and fear in them. A few were concerned whether they could withstand the hot, uncomfortable working environment, wearing their PPE. Only one participant expressed concern about PPE sufficiency. Most student volunteers, including lecturers, spoke of the confidence they had in their local healthcare system in ensuring safety measures were in place (e.g., decontamination procedures, protocols). Thus, concerns relating to volunteer training were afterthoughts.


*“The last factor would be my trust in the system, knowing that your PPE is ALL there...”*
(FGD2, P1, volunteer)


*“Everyone was just going in the dark, and we were very much confident in the higher authority, like they will guide us.”*
(FGD3, P2, lecturer)


*“Maybe because of our status as students we are not very clear on the decon procedures that hospital staff might be familiar with. So that kind of affects the confidence if I were to volunteer… what are the decontamination procedures afterward that (would) prevent me (from) bringing the virus out.”*
(FGD6, P3, non-volunteer)

Protecting their family was one of the primary factors driving students’ decisions not to volunteer. Many revealed that ensuring their family members’ safety outweighed their professional duties of volunteering. Volunteering at the front line meant that they would be exposed to COVID-19 and compromise the safety of their loved ones, particularly the very young and old family members who were deemed more vulnerable. As family held an important place, obtaining parental approval and encouragement contributed to their decision-making. Participants who volunteered said that their parents were supportive, and they would take additional precautionary measures (e.g., showering before going home, and minimizing mingling with family and friends). Students who did not volunteer shared about their parents’ disapproval, and many would not *“go against their wishes”*.


*“I did think of if I insisted on volunteering even though my family wasn’t really keen. Then, what (are the) consequences if I end up contracting the virus? And what kind of burden will (I) put onto the family, like financially and emotionally? So in the end, I just say ‘oh okay, maybe there will be other opportunities in the future to learn… I shouldn’t be risking everything else just for my own learning.”*
(FGD6, P5, non-volunteer)

Peer influence also contributed to some participants’ decision to volunteer. A few sought advice from classmates and seniors; others shared that volunteering with friends *“made the experience more fun”*. Being academically motivated, they contemplated their abilities to manage time. They wanted to fulfill their student role by performing well and completing their studies. While many participants acknowledged that the monetary incentive was highly attractive, it was not *“the biggest pushing factor although it was a key factor”* to account for the health risks involved. A few participants wanted to be accountable to their scholarship/sponsorship providers and were initially unsure whether they could both volunteer and receive monetary incentives.

### 3.2. Bringing Out “the Nurse” in Students through Volunteering

The second theme highlighted nursing students’ enriched volunteering experiences and how the pandemic brought out the “nurse” identity among student volunteers and non-volunteers. Compared with employed registered nurses (RNs), these participants felt entangled and constrained by their “student status” in contributing as a nurse.

#### 3.2.1. Displaying Personal Growth as a Nurse

Student participants who volunteered shared that they gained confidence and proficiency in phlebotomy management. This included technical skill competency, overcoming communication barriers with migrant workers, and observing clinical safety (e.g., patient identification, handling of sharps). Likewise, the lecturers witnessed their clinical progression and were impressed that *“by the end of the sessions, they have already mastered”* phlebotomy care management.

Student volunteers said they gained experiential knowledge about pandemic management and workflow processes. By observing the operation and organization of different zone areas in the dormitories, participants appreciated public health measures and how the local healthcare system functioned during a crisis. They recognized the importance of smooth workflow processes, workspace knowledge, protocol adherence and having buddy systems. Some were more cognizant and appreciative of the roles which laboratory technicians and hospital administrators played. Many participants spoke about the teamwork, camaraderie, and solidarity experienced within the healthcare team.


*“… they (the students) saw how people from the lab... work in a pair…they (the students) saw this kind of coordinated movements and tried to mimic…because every time if you want to prepare to do the things, it takes time. So, one person does passes (of requisites), the second person (does) the withdrawal (to) collect blood. Then you swap the job or roles. So that was also a very good thing they observed and learned.”*
(FGD3, P2, lecturer)


*“It was really very encouraging to see how everyone put aside their roles and positions to come and work together… they didn’t care like how many doctors they have, how many nurses… senior nurses, how many students we have, as long as they meet the manpower needed… everyone just chip in and it doesn’t matter where you come from or your experience. It just matters you contribute.”*
(FGD 4, P4, volunteer)

Student volunteers shared how they grew as a person through frontline volunteering. They were proud that they *“took the leap of faith”*, *“stepped out of their comfort zone”*, and proved to themselves that they could do the job; others learned to troubleshoot problems, built confidence in functioning independently and managing high risks. One participant added, and others agreed, that he learned an important skill—to *“say sorry”*.


*‘I learned… to say sorry. It is truly a vital skill to be very frank and to say that I am new, and I accidentally screwed up or I didn’t manage to take blood. Would you allow me another try?”*
(FGD 4, P2, volunteer)

Student participants mentioned how they learned and displayed the professional attributes of nurses. Through the pandemic, they saw how adaptable nurses are in performing *“multiple roles in multiple situations”*, beyond the clinical environment. Participants learned about the expanded scope of nurses and realized that *“nurses do even more”*. While student participants developed such insights, lecturers observed their spontaneity, passion, and caring demeanor. One student volunteer said, *“Caring is at the heart of what we do”*, and many learned to be *“unafraid to approach the situation”* and *“make the best decisions at that point in time”*. Others shared how they developed resilience, grit, and empathy.

#### 3.2.2. Ascertaining an Identity as a Nurse

Student participants felt a greater sense of belonging to the nursing community after volunteering. They expressed positive feelings toward being able to contribute and enjoy meaningful interactions with migrant workers. Using nursing skills during the pandemic situation made *“us feel like we’re different from the rest”*. They felt recognized for their efforts and were proud to be *“on par in terms of contribution like what the healthcare community was doing for Singapore”*. One participant shared that the staff *“valued me as part of the team though I am a novice to them”*. Those who volunteered unanimously said they would volunteer again. Conversely, those who did not volunteer felt left out, guilty, or regretted not being *“part of the whole team in fighting against COVID-19”*.


*“I would volunteer again. Because I felt like I played my part and I want to play my part in the situation.”*
(FGD4, P1, volunteer)


*“… they had a lack of manpower and that affected their burden at work, so I felt that if I had volunteered, … my conscience (would have) felt a bit better.”*
(FGD6, P3, non-volunteer)

Regardless of their “volunteer status”, the pandemic drew attention to student participants’ sense of professional identity. It not only raised participants’ awareness toward the public’s mixed perception about nurses but also highlighted the societal value that the country placed on nurses. One participant shared how the pandemic strengthened her beliefs on what a nurse does and can do, solidifying her identity as a nurse. Some agreed that the volunteering opportunity broadened their perceptions toward the roles and responsibilities of nurses. Others mentioned that there was no change in how they view nursing, as they already knew what they *“signed up for”*.


*“(The) pandemic wasn’t really in the picture when I was considering to be a nurse... but now the pandemic becomes one of the things I have to overcome as a nurse…I will be willing to learn more about emergency preparedness.”*
(FGD6, P2, non-volunteer)


*“We should have also considered and known that with this kind of education comes a certain responsibility.”*
(FGF4, P2, volunteer)

Most participants affirmed that the pandemic cemented their decision of becoming a nurse and validated their choice of study in the university.


*“It makes me feel like this is…what I can, what I like to do, and what I can foresee myself to do in the future... Like (if) you can go through these very bad times, you can go through more in the future.”*
(FGD2, P5, volunteer)

#### 3.2.3. Entangled in the Student Role: “We Are Not Full Nurses Yet”

Most student participants felt that full-fledged RNs have a duty of care during health crises, as nurses are equipped with specialized healthcare skills to step up. A few were conflicted that nurses had a personal choice to step up, as each had their own concerns. They shared that nurses who did not join the front line should not be viewed as *“wrong”*, *“not courageous enough”* or *“not worthy to be one”*.


*“The public and nation trust us, nurses and healthcare professionals to step up and use their specialized knowledge to help out. And, to add on, they are getting paid. But then, as humans we have every right to our lives…If patients have autonomy, I think nurses and doctors should have autonomy too... basically, knowledge is power. Power is like you can use it or you don’t. It is not an obligation.”*
(FGD 1, P1, volunteer)

Student participants perceived that the professional obligation for RNs to step up is more prominent compared to that of nursing students. As students, they felt they should be given a choice to volunteer. Their priority was to fulfill academic requirements and learn as much as possible so that they could be competent in the future. Many wanted to contribute to the healthcare workforce but were limited by their *“student status”*; they knew they lacked clinical experience and could only perform basic tasks. One participant broached the topic that *“even in (the) clinical attachment, we were highly discouraged from going into isolation rooms... how can we be confident to attend to a pandemic crisis, to such an infectious virus?”* Compared to formal hospital staff, they were not familiar with hospital protocols and workflow. Thus, one participant said it was harder to volunteer as students.

Student participants looked up to and looked forward to being RNs. They highlighted that *“we are not really full nurses yet”* and were *“not fully part of the healthcare family”*. They lacked the RN license and were not employed. A few were concerned about their professional accountability as student volunteers. This added complexity, as they recognized that the schools would have to be accountable for them. Some participants shared that they would be more confident clinically if they were RNs. While they had the choice of volunteering as students, healthcare staff have work responsibilities. Thus, some mentioned it would be easier to convince their parents to allow them to volunteer as it would be *“part of their job”*. A number of participants hoped that they had the financial capability of employed nurses to find alternative accommodation, so that they could go ahead and volunteer, and not worry about infecting their family members.

### 3.3. Gearing Up to Volunteer

The third theme revealed the types and avenues of volunteering opportunities that nursing students could take. It also shed light on how operational workflows can be better managed for future volunteering roles during pandemics.

#### 3.3.1. Healthcare and Non-Healthcare Volunteering Opportunities

Most participants felt that nursing students were equipped with fundamental PPE and infection control competencies to volunteer at the front line, and they *“have the potential to help a lot”*. A few participants expressed strongly that the deployment of healthcare students should precede lay individuals when outsourcing manpower during health crises. Depending on the care demands and clinical expertise needed, participants felt they could be trained and *“learned on the job”*. One student participant highlighted that *“even frontline workers might not be very prepared for this”*, and what was important was to quickly adapt and train individuals for the needed roles. Volunteer training instilled confidence and gave reassurance, by revising the required essential skills.

Alternative avenues of volunteering opportunities were shared. Prior to the university’s invitation, some students expressed interest in contributing in COVID-19 related operations via the Singapore Healthcare Corps, an initiative coordinated by the Singapore Ministry of Health to provide support to healthcare and community care professionals by referring additional healthcare professionals and lay extenders to areas of need in COVID-19 operations and in community care. However, responses were slow, and participants who registered received no updates once the vacancies for volunteering opportunities were taken up. One participant joined a private home nursing agency and conducted health assessments for migrant workers living in dormitories.

Participants also shared alternative volunteering roles in which nursing students could participate. Most of them preferred to volunteer for familiar clinical-related roles that enabled them to apply their nursing knowledge. These included COVID-19 swab tests, basic ADL care and basic clinical assessment, so that hospital nurses could take on advanced responsibilities. A few suggested extending their clinical placement hours in hospitals to increase manpower. Some participants were less enthusiastic in volunteering for administrative and operational support roles. They perceived these roles as less meaningful or lacking interaction with people. On the contrary, a few highlighted that the job scope of such roles could be specified with greater clarity, as they included critical and meaningful work such as patient registration, transporting equipment, and ensuring colleagues gown and de-gown PPE properly. Options could be offered to volunteer in areas having no contact with potential or infected patients. This could ease students’ concerns regarding the risks of infecting themselves and their family members.


*“If there was an option of the logistics one, I might have gone. Because it’s really the direct contact that I was worried about, especially with parents who are quite elderly and vulnerable.”*
(FGD5, P1, non-volunteer)

Others suggested community-based volunteering opportunities to assist vulnerable populations. They included delivering food to the elderly living alone, befriending the elderly, distributing masks, befriending migrant workers via hotlines, and translating health education materials for migrant workers.

#### 3.3.2. Operational Workflow in Managing Student Volunteers

Participants highlighted the inadequate details of the volunteer program in the disseminated recruitment email. They were unsure about the commitment duration, on-site working conditions, decontamination process, presence of clinical supervision on-site, and whether their participation would affect their studies and graduation. Such information was only shared by lecturers during the venipuncture training. Some felt that they were kept *“in suspense”* for too long; the lack of such vital information made them feel less reassured. Some of their peers who registered were put off by the stipulated long volunteering hours, early reporting time and inconvenient volunteering sites. As such, a few withdrew. One participant suggested having a platform or point of contact for interested students to raise inquiries and assist them in making informed decisions. Information delivery could have been more organized and laid out more clearly.


*“Initially when they send out the email, … the commitment is 5 days a week, 7 h a day… it seemed more daunting. But when I hear from my friend’s experiences, it’s not that frequent...”*
(FGD 7, P3, non-volunteer)

Some participants felt the volunteering experiences were brief and wanted to be contacted earlier or allocated more slots so that they could contribute more. The recruitment and training of volunteers could be expedited. Some missed other volunteering opportunities. One participant attributed these negative experiences to the lack of a human resources coordinator to manage volunteers and coordinate volunteering activities.

Nonetheless, a few participants appreciated the opportunity provided by the university; they did not have to manage the administrative procedures of volunteering or worry about getting back to classes on time. Others mentioned that the university *“has done as much as they could”* to transition them to frontline phlebotomists. This included venipuncture training, certification of skill competency, coordinating volunteer schedules, lecturers overseeing the on-site safety of student volunteers, providing psychosocial assurance and advice on risk management and self-care. Having lecturers on-site gave student volunteers a sense of familiarity. They were figures of authority whom students could approach for assistance. Participants gave feedback that apart from having preceptors on-site, help was also constantly available from the ground staff.


*“They were also very kind. If we had any difficulties locating the vein or taking the blood, we could approach any of the experienced people and they would do it for us very willingly.”*
(FGD 2, P4, volunteer)

Participants shared strategies to improve the uptake of future volunteering programs. A few suggested extending the program to first- and second-year nursing students, as well as students from other nursing schools, to increase volunteers. To motivate future student cohorts to volunteer, lecturers suggested producing a video montage to showcase students’ volunteering experiences and inform prospective students about such volunteering opportunities. One lecturer suggested appointing student volunteers as pandemic ambassadors, to give talks and share their experiences with their peers. A few students recommended including pandemic management in the nursing curriculum, using COVID-19 as a case study. While such knowledge could prepare prospective students to be pandemic-ready, some non-volunteers cautioned that such knowledge would not change their decision to volunteer. Most non-volunteers would only change their mind if alternative accommodation were provided, as the risk of infecting family members was their key concern. A few participants suggested counting volunteering time as clinical placement hours to incentivize students to join.

P3 (FGD 4): *“Moving forward, … instead of having the clinical (placement), (as) in like the clean cases, like maybe in the wards, why don’t we give another option for students to clock in clinical hours… during the pandemic…?”*

Moderator: *“… That means the volunteer(ing) to be counted as clinical hours?”*

P3 (FGD 4): *“Yes. Because at least we have the best I would say, best of both worlds (of) being in the clean zone.”*

Participants shared strategies to facilitate the smooth operation of this volunteer program and improve volunteering experiences. Some suggested providing orientation at the dormitories so that they could navigate the operational workflow processes to fit in better. Some participants wished they could have venipuncture practice on one another instead of using manikins. One participant suggested combining low-fidelity skills training with simulation to better manage patients. A few wanted to have more training sessions and an assessment on skill competency to boost their confidence. Others hoped to have more direct and closer on-site supervision for their initial venipuncture attempts.

When asked about what sustained volunteering efforts, participants identified interest, monetary incentives, peer support, camaraderie, assurance, and ease of getting help from approachable faculty members and healthcare staff. Two participants shared that following up with patients’ serological results would allow them to see the impact they were making and sustain their motivation.

## 4. Discussion

The current study revealed the perceptions of final-year pre-registered nursing students volunteering as frontline healthcare workers during the COVID-19 pandemic, focusing on the factors influencing their decision to volunteer, the role of their professional identity as nurses to volunteer, and strategies to improve future volunteering uptake and processes. While the exploration of healthcare students’ perceptions and experiences of volunteering during the COVID-19 pandemic is not new [8,9,14,15], this study contributes to the current literature by demonstrating what worked well, and the efforts to improve future operational processes when recruiting students as frontline healthcare workers to ease manpower constraints.

Our findings suggested that final-year nursing students’ decisions to volunteer as frontline healthcare workers were multifaceted. We observed the interplay of intrinsic and extrinsic factors, as many faced a quandary in their decision-making. As is similar to past studies in COVID-19 and other pandemic situations [10,16,17], most of our student participants were willing and motivated to volunteer. Their motivation was accounted for by Clary and Snyder [18] as six personal and social functions that can be applied to other volunteering contexts in various crisis situations: (1) increasing expression of values such as altruism, (2) seeking learning opportunities and experiences to understand world-views during health crises, (3) enhancing personal growth and psychological development, such as through the fulfillment of their calling and passion pursuit, (4) gaining career-related clinical skills and experiences, (5) fortifying social relationships with peers and beneficiaries (e.g., migrant workers), and (6) protecting oneself from feeling bored, purposeless and guilty for not helping. As is consistent with observations made by our lecturers, Bazan et al. [19] reported that Polish medical students characterized by curious, sensitive, calm, and sociable personalities were more likely to volunteer. Individuals with such traits, linked to extraversion, agreeableness, and openness, have more salient helping identities, and are associated with a greater propensity to volunteer [20].

Protecting family safety emerged as a priority among student participants in this study and was a key determinant of the participants’ decision to not volunteer. Although the fear of transmitting COVID-19 to susceptible significant others was reported among medical and nursing students in Spain, it did not stop them from volunteering [14]. Our findings highlighted the strong collectivist Asian family values and culture that Singapore students hold, regarding placing their family first. Similar findings on the fears for their family’s health were reported among medical students in Indonesia [10]. Family interests are expected to overrule those of the individual [21]. As such, the provision of alternative accommodation for these non-volunteers would change their minds in stepping forward and increase the volunteering uptake. Offering concessionary prices for accommodation within the university campus could be a possible approach to support these students [15]. During the COVID-19 pandemic, non-volunteer nursing students in our study displayed the citizenship responsibilities of protecting family members, instead of assuming the professional duties of a nurse [8].

Despite this uncertainty, it was encouraging to learn that our participants had confidence in Singapore’s healthcare system to enforce safety and operational measures for healthcare workers and volunteers. Some students were thus reassured enough to volunteer without worrying about training in PPE and venipuncture. This takes into consideration that our nursing students were approached in May 2020, when hundreds of new COVID-19 cases were reported daily in migrant-worker dormitories. Their confidence in Singapore’s healthcare system could be attributed to the establishment of existing pandemic workflows and mitigation processes. Subsequently, when the participants were interviewed between September 2020 and November 2020, the spread of COVID-19 was under control and there were no migrant-worker cases detected. Our findings identified the importance of student volunteers having confidence in the healthcare system. During the earlier phases of the COVID-19 pandemic, healthcare student volunteers in other countries reported safety concerns related to insufficiency and limited access to PPE, and the lack of knowledge towards equipment usage [14,19].

Our study participants acknowledged that the monetary remuneration enticed them to volunteer and sustained their volunteering efforts. This differed from Lazarus et al.’s study [10], which reported a lack of association between monetary incentives and increased willingness to volunteer among undergraduate medical students in Indonesia. Additionally, our student participants suggested including volunteering as part of their clinical placement hours to incentivize peers to volunteer. Such academic-related incentives, which some healthcare education institutes in the United States and Europe employed as part of service-learning, internship programs, or curricular activities, require a prompt responsive curriculum redesign and strong hospital/community–campus partnerships [15,22,23]. While such formal or even mandatory academic incentives can encourage volunteer participation, they might reduce students’ internal motivation and satisfaction regarding volunteer work [15]. To mitigate the negative effects and sustain volunteering efforts, as revealed in our findings, autonomy to choose the type and location of volunteer work could be given to students, as well as increasing students’ perception of intrinsic motivation by demonstrating how volunteer work fits various goals [24,25].

The pandemic reminded student participants of the professional function of the healthcare knowledge and capabilities they were developing, which highlighted and strengthened their professional identity as nurses. While student volunteers had experiential learning, non-volunteers gained such perspectives through observations and peer-sharing. Nursing students generally related the “nurse” identity with their clinical abilities and gained new insights on the application of nursing skills [26]. At the personal level, our study participants displayed their “nurse” identity through acts of assertiveness, compassion, competence, confidence, conscience, commitment, and courage during the pandemic [27]. At the interpersonal level, they experienced continual personal growth, maturity and affirmation through professional socialization, as they cultivated their sense of belonging to the nursing community [27]. At the societal level, pandemics such as COVID-19 provided students with an opportunity to develop the psyche for forming an evolved professional identity of being “system citizens” who contribute to the needs of the healthcare system [22].

However, our findings showed that student participants’ professional identity as nurses was still solidifying, which likely accounted for their wavering thoughts on volunteering. Toggling with their student role status, some participants expressed fear and uncertainty regarding their academic progress. This was also evident among final-year nursing students in Spain who were forced to work before they completed their studies [8]. Nonetheless, we observed a presenting range of professional maturity as nurses among the student volunteers. While some volunteers were in the “learner” mode of wanting to gain more skills and training, others transcended to the “professional” realm of volunteering dutifully, with the intention of serving as a nurse. As such, our study participants felt that they were “not full nurses yet” when volunteering as phlebotomists. This contrasted with Gomez-Ibanez et al.’s study [8], where final-year nursing students perceived themselves as nurses when they were pushed to mature faster professionally and function as nurses in the workforce. Volunteer administrators and educators thus need to be mindful and support such a transitionary phase of professional development among healthcare student volunteers. Examples of approaches identified were validating and addressing their academic concerns, providing adequate competency volunteer training and assessment, and empowering students to perform related clinical volunteer work independently and safely, to bridge the gap between being students and “professional” volunteers.

Student volunteers’ sense of belonging to the profession was cemented as they accumulated clinical experiences through volunteering, and received consistent support and recognition from healthcare staff and the general public. Our findings supported past studies that feeling welcomed by healthcare staff, amicable staff interactions and behaviors, team comradeship and receiving appreciation are components instrumental in positive staff-student relationships, and cultivate a sense of belonging to sustain volunteering efforts [26,28]; these approaches made students feel valued and closer to being a member of the profession [26]. Thus, promoting cohesive and positive staff–student volunteer relationships, facilitating clinical guidance by allocating preceptors, providing psychosocial support, and close follow-up with students’ interactions with healthcare staff are crucial in sustaining students’ volunteering commitment and efforts.

Our student participants shared a wealth of alternative volunteering roles which they can contribute as part of their participatory service learning. Other studies also articulated roles in promoting pandemic safety awareness and mitigating the consequences of pandemic safety measures (e.g., social isolation) in the community, addressing pandemic-specific and non-COVID-19 healthcare and hospital operational processes, supporting daily living concerns outside the work responsibilities (e.g., childcare) of healthcare staff, and assisting in COVID-19-related research work [22,23,29]. In Singapore, we have a relatively large pool of more than 6000 nursing students, with various years of study, across different educational institutions. Identifying these potential areas of volunteering allows healthcare educational institutions, hospitals, and community agencies to tap students’ capabilities and capacities for future pandemics.

Our findings highlighted the need to establish a tight volunteer management team from the university to ensure collaboration, coordination and regularly updated communication with our hospital/community partners and students. Such a team serves as a bridge between the university and hospital/community partners, attends to the details of operational processes, and facilitates smoother volunteer recruitment, training, placement, and coordination. In addition to articulating well-defined roles and clear workflows, Long et al. [22] emphasized that the first action of such a team was to ensure regulatory and safety measures were in place within the legal parameters and students’ capacity. Depending on the nature of the voluntary tasks, contact proximity, and duration of contact with at-risk or infected COVID-19 patients, students may incur physical or psychological harm to themselves or others, and subject involved stakeholders to liability risks arising from their participation in spontaneous volunteering [3]. Fortunately, no needlestick injury was reported and our student volunteers were swabbed as negative for COVID-19 at the end of the program.

For healthcare students to function effectively in clinical-related volunteering roles, they need to be equipped with core competencies and well-prepared for pandemic or emergency situations [10]. Our findings revealed that students were not familiar with how healthcare systems respond during health emergencies. The lack of such knowledge was reported to be a barrier to volunteering during the COVID-19 pandemic among undergraduate healthcare students in Saudi Arabia [9]. Despite previous historical encounters with SARS, MERS and swine flu, not many universities, including our own, embed pandemic preparedness and education on the logistical challenges specific to pandemics in the undergraduate curriculum to ensure prospective healthcare workers are adequately prepared for future public health emergencies [30]. While our findings revealed that knowledge on pandemic preparedness might not increase the volunteering uptake, it could increase prospective healthcare professionals’ reception, willingness, readiness, and confidence in managing future health emergencies. Our findings also shared strategies for healthcare educational institutions to showcase and encourage volunteerism among students during health emergencies.

### Strengths and Limitations

One key strength of this study is the participation of nursing student volunteers and non-volunteers, as well as all the faculty involved in supervising the student volunteers on-site. The inclusion of different stakeholders for data source triangulation captured wide-ranging perspectives and allowed data validation [31]. Additionally, investigator triangulation via independent coding was performed by two researchers, followed by extensive discussion of results with other team members to enhance trustworthiness [31].

One limitation of this study is that considering the impact of the COVID-19 outbreak, e.g., incidence and mortality rate, and how Singapore’s response to the virus differed from other countries, the findings of this study might be context-specific and might not be generalizable to other nursing student populations. The second limitation of this study was the small sample size of faculty members included, as there were only three faculty members who supervised or worked alongside the student volunteers. Another study limitation that might affect the generalizability of this study is the lack of ethnic representation among the nursing student volunteers sampled.

## 5. Conclusions

Our study affirmed that pre-registered nursing students are uniquely positioned to contribute gainfully during pandemics. These nursing students brought their health knowledge, awareness of the delivery of healthcare, understanding of COVID-19 as a societal challenge, professional maturity, and commitment to serve healthcare needs [22]. They should be considered as contingent resources in times of healthcare emergency crises, and they have the capacity and capability to take on both frontline and supportive roles to ease the burden on the healthcare workforce and the cost on society. Thus, our findings advocate strategic partnerships between the hospital/communities and academic institutions in providing various healthcare services during pandemic crises.

Our findings highlighted the importance of addressing pre-registered nursing students’ extrinsic concerns to improve the volunteering uptake. Strategies identified included providing accommodation, as well as monetary and academic-related incentives. From the education and clinical perspectives, emergency preparedness and the scope of professional responsibilities during a pandemic should be introduced to the curriculum of pre-registered nursing students, to cultivate the “readiness to serve” attitude and aptitude in times of healthcare emergencies. Additionally, supporting the transitionary phase from students to “professional volunteers”, promoting cohesive and positive staff-student volunteer relationships, and establishing volunteer management teams were strategies that were identified to improve operational processes when engaging pre-registered nursing students as frontline workers.

In summary, our findings suggested that we must look beyond the simplicity of altruism to the role of professional identity, operational, and motivational factors to explain final-year nursing students’ intentions to volunteer and their volunteer behavior.

## Figures and Tables

**Table 1 ijerph-18-06668-t001:** Characteristics of study participants (*n* = 33).

Characteristic	Nursing Students		Lecturers (*n* = 3)
	Volunteers (*n* = 15)	Non-Volunteers (*n* = 15)	
Gender			
Female	9	13	2
Male	6	2	1
Age (Mean ± SD)	23.5 ± 2.0	22.6 ± 0.9	-
Ethnicity			
Chinese	15	12	1
Malay	0	1	1
Indian	0	2	0
Other	0	0	1
Previous volunteering experience			
Yes	14	9	-
No	1	5	-
Prefer not to say	0	1	-

**Table 2 ijerph-18-06668-t002:** Semi-structured interview guide.

Willingness of final-year nursing students to volunteer as frontline healthcare personnel during the COVID-19 pandemic (volunteers/non-volunteers/lecturers) - Do you think healthcare students should be deployed to be frontline healthcare personnel during pandemic crises or national emergencies? Why? - Do you think final-year nursing students are equipped to volunteer as frontline healthcare personnel? Why? - Do you think if we give training to nursing students, they would be expected to volunteer? Why? - What are some of the volunteering activities that healthcare students can contribute? Nursing students’ consideration factors for volunteering or not volunteering as frontline healthcare personnel (volunteers and non-volunteers) - What are some of the considerations to determine if you will volunteer? - How did you come to a decision to volunteer/not to volunteer as a frontline phlebotomist? - What are the top three key factors that led you to make that decision to volunteer/not volunteer? - Was it an easy or difficult decision to come to? Why? Role of professional identity in influencing the decision and experience of volunteering as frontline healthcare personnel - Do you think nursing students are professionally obligated/have a duty of care to volunteer in times of pandemic crisis? Why? (volunteers/non-volunteers/lecturers) - Do you think the students’ professional identity was demonstrated during the course of their volunteering? Why? (lecturers) - Has your volunteering experience (volunteers)/the pandemic experience (non-volunteers) changed the way you view yourself as a nurse? In what way? - How was your experience of volunteering? (volunteers) - If you were asked to volunteer again, will you do it again? Why? (volunteers) - Do you feel part of the healthcare community? (volunteers/non-volunteers) Areas of improvement to increase voluntary participation as frontline healthcare personnel and to enhance the volunteering experience - What do you hope to change if you were to volunteer again? (volunteers) - What would discourage you to volunteer again? (volunteers) - What will make you change your mind to volunteer? (non-volunteers) - What would encourage nursing students to volunteer during a pandemic (lecturers)

**Table 3 ijerph-18-06668-t003:** Process of data analysis.

Theme	Sub-Theme	Reduced Codes
Wavering thoughts on volunteering	Propelling intrinsic motivators	To help Concern for migrant workers Personality Apply knowledge and learn skills Seek experiences Have something to do Not joining healthcare post-graduation
	Accounting for extrinsic concerns	Pandemic uncertainty Confidence in the healthcare system Protecting family members Peer influence Academic concerns Attractive incentives Accountable to scholarship/sponsorship providers
Bringing out “the nurse” in students through volunteering	Displaying personal growth as a nurse	Gaining confidence and proficiency in phlebotomy management Appreciating pandemic management and workflow Growing as a person Demonstrating the attributes of a nurse
	Ascertaining the identity as a nurse	Being part of the nursing community through volunteering Strengthening the perceived identity of a nurse Affirming the decision to be a nurse
	Entangled in the student role: “we are not full nurses yet”	Duty of care as healthcare professionals Student status Student nurse versus employed nurse
Gearing up to volunteer	Healthcare and non-healthcare volunteering opportunities	Healthcare students are equipped to volunteer Ways and processes of signing up as a volunteer Types of volunteering opportunities
	Operational workflow in managing student volunteers	Organizing and providing details of the volunteering task Providing venipuncture training for the volunteering task Help is available when needed Strategies to promote the uptake of volunteering Strategies to promote the sustainability of volunteering

## Data Availability

Raw data on the focus group discussion transcripts could be made available upon request. The data are not publicly available due to privacy.

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
