# Peer review of "To Volunteer or Not? Perspectives towards Pre-Registered Nursing Students Volunteering Frontline during COVID-19 Pandemic to Ease Healthcare Workforce: A Qualitative Study"

_ijerph, 2021, doi:10.3390/ijerph18126668_

Round 1
Reviewer 1 Report
The manuscript entitled “To volunteer or not? Perspectives towards pre-registered nursing students volunteering frontline during COVID-19 pandemic to ease healthcare workforce: a qualitative study” presents interesting issue, but some areas must be corrected.
Major:
- Authors prepared interviews in 3 groups: students who volunteered (n=15), students who did not volunteer (n=15) and lecturers who supervised student volunteers (n=3). A really major problem (especially for a group of lecturers who supervised student volunteers) is associated with a small number of interviews (3 respondents per group for lecturers), while it is in general recommended to conduct at least 20 interviews per group to obtain a proper sample size in qualitative research. Authors should get familiar with the results of the systematic analysis of qualitative health research over a 15-year period by Vasileiou et al. (https://www.ncbi.nlm.nih.gov/pmc/articles/PMC6249736/). Without a proper sample size, the obtained results do not allow to conclude, as they are seriously biased. Authors should gather more respondents to properly conclude. Authors should include more respondents in a groups of students and lecturers, or include them only in a group of students and remove presentation of data based on the interviews with lecturers as in the group of 3 persons they can not formulate any valid observations.
- There is a problem with ethnicity of gathered respondents. While Singapore is an multiethnic country, Authors gathering a group of students who volunteered (n=15), gathered only Chinese ethnicity. As a result, Authors do not observe a broad spectrum of experiences which may be associated with ethnicity. Moreover, it may cause false conclusions (e.g. impression that among volunteers were only Chinese students), which may create false beliefs and misunderstanding.
- Authors did not present their results in the context of the situation in Singapore. Reader even does not know what was the number of COVID-19 cases in Singapore, what legal regulations associated with COVID-19 were in charge, what was the public health policy, etc. Authors should present situation and discuss their results in the context of it. While presenting situation, Authors should focus on 2 periods: (1) period when students volunteered (as it may have influenced their decision to do it), (2) period when the study was conducted (as it may have influenced their perception and answers that they provided).
- Authors should broaden the section presenting the results of their study, as they chosen only some statements and it seems that they based their description of the results on only single statements. Authors should present more statements being the bases of their description, in order to present properly the scope of opinions which were obtained. Such presentation which Authors chosen causes that we may suppose that they present only those results which they want to be in agreement with their statements.
- Authors should broaden their discussion to present also the general approach associated with volunteering, not only volunteering of nursing students during COVID-19, but also the general motives to volunteer in various crisis situations.
- Authors should formulate more specific conclusions which are based on the conducted study and present any real observations. Statements such as “Findings suggested that we must look beyond the simplicity of altruism to the role of professional identity, operational, and motivational factors to explain final year nursing students’ intention to volunteer and their volunteer behavior.” does not present any “real” observation.
Author Response
Dear reviewer,
Thank you for providing your constructive feedback. Below are my responses:
- Authors prepared interviews in 3 groups: students who volunteered (n=15), students who did not volunteer (n=15) and lecturers who supervised student volunteers (n=3). A really major problem (especially for a group of lecturers who supervised student volunteers) is associated with a small number of interviews (3 respondents per group for lecturers), while it is in general recommended to conduct at least 20 interviews per group to obtain a proper sample size in qualitative research. Authors should get familiar with the results of the systematic analysis of qualitative health research over a 15-year period by Vasileiou et al.
(https://www.ncbi.nlm.nih.gov/pmc/articles/PMC6249736/). Without a proper sample size, the obtained results do not allow to conclude, as they are seriously biased. Authors should gather more respondents to properly conclude. Authors should include more respondents in a groups of students and lecturers, or include them only in a group of students and remove presentation of data based on the interviews with lecturers as in the group of 3 persons they can not formulate any valid observations.
Response: As we only had 3 lecturers who supervised the 27 student volunteers, all of them were recruited in this study. The intent of recruiting the lecturers was to provide triangulation of data on the volunteering experiences of students and gather recommendations from them on how the volunteering uptake can be enhanced for future similar initiatives. The authors are aware that the sample size of 3 respondents for lecturers is small and acknowledged that this is a study limitation that impacts the validity/generalisability of the study. As such, we have included the following sentences
‘and three lecturers who provided…” under section 2.1; and “Although our study included a small sample size of faculty members…” to acknowledge this limitation under section 4.1.
- There is a problem with ethnicity of gathered respondents. While Singapore is an multiethnic country, Authors gathering a group of students who volunteered (n=15), gathered only Chinese ethnicity. As a result, Authors do not observe a broad spectrum of experiences which may be associated with ethnicity. Moreover, it may cause false conclusions (e.g. impression that among volunteers were only Chinese students), which may create false beliefs and misunderstanding.
Response: Thank you for highlighting that. We have included the following sentence in section 4.1 to acknowledge this limitation: “Other study limitation which might affect the generalizability of this study is the lack of ethnicity representation among the nursing student volunteers sampled.”
- Authors did not present their results in the context of the situation in Singapore. Reader even does not know what was the number of COVID-19 cases in Singapore, what legal regulations associated with COVID-19 were in charge, what was the public health policy, etc. Authors should present situation and discuss their results in the context of it. While presenting situation, Authors should focus on 2 periods: (1) period when students volunteered (as it may have influenced their decision to do it), (2) period when the study was conducted (as it may have influenced their perception and answers that they provided).
Response: The following sentence was included in the background to provide the number of COVID-19 cases per day:“Hundreds of new migrant worker cases were detected each day with numbers soaring to a peak of 1,397 on 20 April 2020.”
We adopted the suggestion of discussing the results in context of Singapore’s situation. The following has been amended in the discussion: “This was in consideration that our nursing students were approached in May 2020, when hundreds of new COVID-19 cases were reported daily in migrant worker dormitories. Their confidence in Singapore’s healthcare system could be attributed by the establishment of existing pandemic workflows and mitigation processes. Subsequently, when the participants were interviewed between September 2020 to November 2020, the spread of COVID-19 was under control and there were no migrant worker cases detected. Our findings identified the importance of student volunteers having confidence in the healthcare system.”
- Authors should broaden the section presenting the results of their study, as they chosen only some statements and it seems that they based their description of the results on only single statements. Authors should present more statements being the bases of their description, in order to present properly the scope of opinions which were obtained. Such presentation which Authors chosen causes that we may suppose that they present only those results which they want to be in agreement with their statements.
Response: Our manuscript adheres to the Braun and Clarke (2006)’s six steps of thematic analysis and this was cited in the methods section. As recommended in the last phase of analysis on report writing, “Choose particularly vivid examples, or extracts which capture the essence of the point you are demonstrating, without unnecessary complexity. The extract should be easily identifiable as an example of the issue. However, your write-up needs to do more than just provide data. Extracts need to be embedded within an analytic narrative that compellingly illustrates the story you are telling about your data, and your analytic narrative needs to go beyond description of the data, and make an argument in relation to your research question (pg.93)”
As such, the quotes used in the manuscript were selected to best demonstrate and justify the description of results.
- Authors should broaden their discussion to present also the general approach associated with volunteering, not only volunteering of nursing students during COVID-19, but also the general motives to volunteer in various crisis situations.
Response: Thank you for the comment. As the aims of this paper were to identify factors influencing willingness/decision of nursing students to volunteer and strategies to improve future volunteering uptake to alleviate the shortage of healthcare manpower, discussions are kept tight to focus on the volunteering of nursing students. General motives to volunteer among individuals may differ across the nature and types of volunteering tasks in times of different crises.
- Authors should formulate more specific conclusions which are based on the conducted study and present any real observations. Statements such as “Findings suggested that we must look beyond the simplicity of altruism to the role of professional identity, operational, and motivational factors to explain final year nursing students’ intention to volunteer and their volunteer behavior.” does not present any “real” observation.
Response: Thank you for the feedback. We have refined our conclusion and present real observations from our study as follows:
‘Our study affirmed that pre-registered nursing students are uniquely positioned to contribute gainfully during pandemics. These nursing students brought their health knowledge, awareness towards delivery of healthcare, understanding of COVID-19 as a societal challenge, professional maturity, and commitment to serve healthcare needs [22]. They should be considered as contingent resources in times of healthcare emergency crises and they have the capacity and capability to take on both frontline and supportive roles to ease the burden of healthcare workforce and on society. Thus, our findings advocate strategic partnerships between hospital/ communities and academic institutions in providing various healthcare services during pandemic crises.
Our findings highlighted the importance of addressing pre-registered nursing students’ extrinsic concerns to improve the volunteering uptake. Strategies identified included providing accommodation, as well as monetary and academic-related incentives. From the education and clinical perspectives, emergency preparedness and scope of professional responsibilities during pandemic should be introduced to the curriculum of pre-registered nursing students to cultivate the ‘readiness to serve’ attitude and aptitude in times of healthcare emergencies. Additionally, supporting transitionary phase from students to ‘professional volunteers’, promoting cohesive and positive staff-student volunteer relationships and establishing volunteer management team were strategies identified to improve operational processes when engaging pre-registered nursing students as frontline workers.
In summary, our findings suggested that we must look beyond the simplicity of altruism to the role of professional identity, operational, and motivational factors to explain final year nursing students’ intention to volunteer and their volunteer behavior.’
Reviewer 2 Report
Please see attachment.

Author Response
Dear reviewer,
Thank you for your positive feedback. Below are our responses:
- Theoretical Framework:
While the Introduction has illustrated the current state of healthcare staff shortage due to COVID-19 and the engagement of healthcare students in volunteering for the frontline, however, as a qualitative study, the authors can benefit from introducing the logical framework to explain the healthcare students’ willingness to volunteer.
Here, I recommend the authors to consult with the mindsponge mechanism (https://www.sciencedirect.com/science/article/abs/pii/S0147176715000826?via%3Dihub
).
The mindsponge mechanism explains how individuals process information and decide, with underlying themes of a multi-layer filter and inductive attitude. As the healthcare students’ willingness and motivation to volunteer has to be based on processing information and making the decision, the mindsponge mechanism can be a strong guideline for the authors and the audiences to navigate the findings.
Results The theoretical framework can help to explain the logic behind the three identified themes. Currently, at the beginning of section 3. Results, it is unclear how the authors identified these three themes. Thus, a clear explanation of the logic behind this identification will provide a deeper understanding of the findings.
Response: Thank you for introducing the mindsponge mechanism as a potential theoretical framework to make sense of the themes derived in the manuscript. We read the suggested paper and found it insightful in understanding how individuals process information. However, our qualitative data was analysed inductively using thematic analysis, without the use of any theoretical framework to identify the themes deductively. Our themes were identified inductively using Braun and Clarke’s six steps of thematic analysis (2006), as described.
Discussion The discussion can benefit from a comparison with findings in other contexts. For instance:
- Saudi Arabia: https://bmjopen.bmj.com/content/11/2/e042910
- Indonesia: https://bmcmededuc.biomedcentral.com/articles/10.1186/s12909-021-02576-0
- Poland: https://www.ncbi.nlm.nih.gov/pmc/articles/PMC7871109/
- Nigeria:https://www.npmj.org/article.asp?issn=1117-1936;year=2021;volume=28;issue=1;spage=1;epage=13;aulast=Adejimi
Different cultures and different COVID-19 situations in different countries will certainly provide a different understanding of the topic. For instance, different family values will have a different impact on healthcare students. In Asia, the author should also consider the phenomenon of cultural additivity, which have proved to be crucial in establishing tactics to deal with COVID-19. You can read the following articles for reference: https://www.frontiersin.org/articles/10.3389/fpsyt.2020.589618/full
https://www.mdpi.com/2071-1050/12/7/2931
Response: Thank you for suggesting various relevant references to enhance the discussion of results. In our discussion, we compared our findings with studies conducted in other contexts, and some of which were studies that you have kindly provided. Perhaps, we were not explicit in our paper and the following parts of the discussion have been amended to illustrate the comparisons.
(Discussion- third paragraph): ‘Consistent with observations made by our lecturers, Bazan et al. [17] reported Polish medical students characterised by curious, sensitive, calm, and sociable personalities were more likely to volunteer.’
(Discussion- fourth paragraph): ‘Similar findings on fears for family’s health were reported among medical students in Indonesia [10].’
(Discussion- sixth paragraph): ‘This differed from Lazarus et al.’s study [10] which reported a lack of association between monetary incentives and increased willingness to volunteer among undergraduate medical students in Indonesia.’
(Discussion- last paragraph): ‘The lack of such knowledge was reported to be a barrier to volunteer during COVID-19 pandemic among undergraduate healthcare students in Saudi Abrabia [9].’
Conclusion…One critical aspect that has been left unanswered is the financial burdens on the patients during the pandemic, due largely to multiple sources of uncertainties occurring in society. And in these unprecedented times, the participation of volunteers as part of the healthcare workforce will not just represent ethical and professional behaviors but also help ease the overall burden on society.
Response: Indeed, financial burdens on patients during the pandemic is a pertinent issue that healthcare systems across the globe faced. We have amended the conclusion
‘… to ease the burden of healthcare workforce and cost on society.” Under conclusion, line 689
Reviewer 3 Report
I appreciate the opportunity to review this article. The issue raised is very interesting, current and necessary for the health crisis of COVID-19. Here are some suggestions for improvement to consider the publication of this manuscript in the journal. In the introduction section, I consider it important to reference bibliographic citations according to the format required by the journal. From line 61 and for two paragraphs there are no appointments. It is important to understand the contextualization and rationale of the study to understand why and the results of the study. The authors describe that informed consent was requested and the study protocol was applied. Does this refer to a favorable report from an ethics committee? I find it interesting that they specify what the proptoclo consists of or that the number of the report obtained is added. The explanatory tables are very useful and I suggest that the authors review the format required by the journal. I congratulate the authors for the writing of the results and discussion sections, they are of great help to understand the conclusions. Likewise, I recommend an in-depth review of the English language.Author Response
I appreciate the opportunity to review this article. The issue raised is very interesting, current and necessary for the health crisis of COVID-19. Here are some suggestions for improvement to consider the publication of this manuscript in the journal.
In the introduction section, I consider it important to reference bibliographic citations according to the format required by the journal. From line 61 and for two paragraphs there are no appointments. It is important to understand the contextualization and rationale of the study to understand why and the results of the study.
Response: Thank you for the highlight. We have added further information to allow readers to have a better understanding of the COVID-19 situation in Singapore.
The following has been added in the introduction:
“In Singapore, the COVID-19 crisis evolved from multiple community cluster outbreaks in February 2020 to large endemic spreads in migrant worker dormitories beginning April 2020. These dormitories were high-density communal residences with unhygienic living conditions for low-waged migrant workers who lacked equal access to healthcare and social safety nets, and these factors elevated risk for large outbreaks of respiratory dis-eases [11]. Hundreds of new migrant worker cases were detected daily with numbers soaring to a peak of 1,397 on 20 April 2020 [12]. A national taskforce was established to coordinate Singapore’s outbreak response.”
The authors describe that informed consent was requested and the study protocol was applied. Does this refer to a favorable report from an ethics committee? I find it interesting that they specify what the proptoclo consists of or that the number of the report obtained is added.
Response: Yes, approval to conduct this study was obtained from the university’s ethics committee. The following sentence has been amended to enhance clarity: “Prior to the start of data collection, the study protocol was approved by the University’s Institutional Review Board ethics committee.”
The explanatory tables are very useful and I suggest that the authors review the format required by the journal. I congratulate the authors for the writing of the results and discussion sections they are of great help to understand the conclusions. Likewise, I recommend an in-depth review of the English language.
Response: Thank you for your positive comment. We reviewed and adhered to the format of the tables as required by the journal. The paper has also been proofread prior to the manuscript submission.
Reviewer 4 Report
Great work. Minor corrections noted within the text and the need for future clinical implications within or as part of the conclusion would be helpful. Great the study has been done, however, what is the take home message for the international audience if they were to do some thing similar?
Other corrections, please see attached
Overall, well done.

Author Response
Great work. Minor corrections noted within the text and the need for future clinical implications within or as part of the conclusion would be helpful. Great the study has been done, however, what is the take home message for the international audience if they were to do something similar?
Response: Thank you for the positive comments and proofreading the manuscript. We have amended the conclusion to include the implications of our study as follows:
‘‘Our study affirmed that pre-registered nursing students are uniquely positioned to contribute gainfully during pandemics. These nursing students brought their health knowledge, awareness towards delivery of healthcare, understanding of COVID-19 as a societal challenge, professional maturity, and commitment to serve healthcare needs [22]. They should be considered as contingent resources in times of healthcare emergency crises and they have the capacity and capability to take on both frontline and supportive roles to ease the burden of healthcare workforce and cost on society. Thus, our findings advocate strategic partnerships between hospital/ communities and academic institutions in providing various healthcare services during pandemic crises.
Our findings highlighted the importance of addressing pre-registered nursing students’ extrinsic concerns to improve the volunteering uptake. Strategies identified included providing accommodation, as well as monetary and academic-related incentives. From the education and clinical perspectives, emergency preparedness and scope of professional responsibilities during pandemic should be introduced to the curriculum of pre-registered nursing students to cultivate the ‘readiness to serve’ attitude and aptitude in times of healthcare emergencies. Additionally, supporting transitionary phase from students to ‘professional volunteers’, promoting cohesive and positive staff-student volunteer relationships and establishing volunteer management team were strategies identified to improve operational processes when engaging pre-registered nursing students as frontline workers.
In summary, our findings suggested that we must look beyond the simplicity of altruism to the role of professional identity, operational, and motivational factors to explain final year nursing students’ intention to volunteer and their volunteer behavior.’
Round 2
Reviewer 1 Report
The manuscript entitled “To volunteer or not? Perspectives towards pre-registered nursing students volunteering frontline during COVID-19 pandemic to ease healthcare workforce: a qualitative study” presents interesting issue, but some areas must be corrected. Unfortunately Authors ignored some of my previous comments, so I presented broadened comments below.
Major:
Authors prepared interviews in 3 groups: students who volunteered (n=15), students who did not volunteer (n=15) and lecturers who supervised student volunteers (n=3). A really major problem (especially for a group of lecturers who supervised student volunteers) is associated with a small number of interviews (3 respondents per group for lecturers), while it is in general recommended to conduct at least 20 interviews per group to obtain a proper sample size in qualitative research. Authors should get familiar with the results of the systematic analysis of qualitative health research over a 15-year period by Vasileiou et al. (https://www.ncbi.nlm.nih.gov/pmc/articles/PMC6249736/). Without a proper sample size, the obtained results do not allow to conclude, as they are seriously biased. Authors should gather more respondents to properly conclude. Authors should include more respondents in a groups of students and lecturers, or include them only in a group of students and remove presentation of data based on the interviews with lecturers as in the group of 3 persons they can not formulate any valid observations.
Authors agreed in their response letter that such sample size is too small and they declared in thei response letter that they indicated it as a limitation. However, the sentence whihc they formulated „Although our study included a small sample size of faculty members, the inclusion of different stakeholders for data source triangulation captured wide-ranging perspectives and allowed data validation” does not present it in any way as a limitation. Authors must present objective assessment of their study while presentation of limitations should be justified and honest. Authors can not formulate their limitations as „advertising” to hide the weaknesses and mislead readers.
Authors should broaden the section presenting the results of their study, as they chosen only some statements and it seems that they based their description of the results on only single statements. Authors should present more statements being the bases of their description, in order to present properly the scope of opinions which were obtained. Such presentation which Authors chosen causes that we may suppose that they present only those results which they want to be in agreement with their statements.
As Authors indicated in their response letter, they should have chosen particularly vivid examples, but there should be more of tchem to deepen the experience of resders.
Authors should broaden their discussion to present also the general approach associated with volunteering, not only volunteering of nursing students during COVID-19, but also the general motives to volunteer in various crisis situations.
In spite of the fact that it was not the aim of the study, the general motives to volunteer in various crisis situations are general background to volunteer in the case of COVID.
Author Response
Dear Reviewer,
Thank you for your comments. Kindly refer to our responses below.
The manuscript entitled “To volunteer or not? Perspectives towards pre-registered nursing students volunteering frontline during COVID-19 pandemic to ease healthcare workforce: a qualitative study” presents interesting issue, but some areas must be corrected. Unfortunately Authors ignored some of my previous comments, so I presented broadened comments below.
Major:
Authors prepared interviews in 3 groups: students who volunteered (n=15), students who did not volunteer (n=15) and lecturers who supervised student volunteers (n=3). A really major problem (especially for a group of lecturers who supervised student volunteers) is associated with a small number of interviews (3 respondents per group for lecturers), while it is in general recommended to conduct at least 20 interviews per group to obtain a proper sample size in qualitative research. Authors should get familiar with the results of the systematic analysis of qualitative health research over a 15-year period by Vasileiou et al. (https://www.ncbi.nlm.nih.gov/pmc/articles/PMC6249736/). Without a proper sample size, the obtained results do not allow to conclude, as they are seriously biased. Authors should gather more respondents to properly conclude. Authors should include more respondents in a groups of students and lecturers, or include them only in a group of students and remove presentation of data based on the interviews with lecturers as in the group of 3 persons they can not formulate any valid observations.
Authors agreed in their response letter that such sample size is too small and they declared in thei response letter that they indicated it as a limitation. However, the sentence whihc they formulated „Although our study included a small sample size of faculty members, the inclusion of different stakeholders for data source triangulation captured wide-ranging perspectives and allowed data validation” does not present it in any way as a limitation. Authors must present objective assessment of their study while presentation of limitations should be justified and honest. Authors can not formulate their limitations as „advertising” to hide the weaknesses and mislead readers.
Response: We have removed the phrase ‘Although our study included a small sample size of faculty members’ and included the following statement, “The second limitation of this study was the small sample size of faculty members included as there were only three faculty members who supervised or worked alongside with the student volunteers.” Future studies should include more faculty participants for generalizability.
Authors should broaden the section presenting the results of their study, as they chosen only some statements and it seems that they based their description of the results on only single statements. Authors should present more statements being the bases of their description, in order to present properly the scope of opinions which were obtained. Such presentation which Authors chosen causes that we may suppose that they present only those results which they want to be in agreement with their statements.
As Authors indicated in their response letter, they should have chosen particularly vivid examples, but there should be more of tchem to deepen the experience of resders.
Response: We included more quotes to the results section of the manuscript to provide different perspectives, so that description of results would not be based on single statements.
Authors should broaden their discussion to present also the general approach associated with volunteering, not only volunteering of nursing students during COVID-19, but also the general motives to volunteer in various crisis situations.
In spite of the fact that it was not the aim of the study, the general motives to volunteer in various crisis situations are general background to volunteer in the case of COVID.
Response: Thank you for the clarification. In fact, we accounted for the general motivations to volunteer in our discussion, citing Clary and Snyder (1999)’s six personal and social functions of volunteering (values, understanding, enhancement, career, social and protective), which could be applied to any population. Perhaps, owning to the way how we applied these six functions of volunteering in our discussion to nursing students, they were not explicitly illustrated. The following sentences have been rephrased to enhance clarity
“Their motivation were accounted by Clary and Snyder [18] six personal and social functions which can be applied to other volunteering contexts in various crisis situations: (1) increasing expression of values such as altruism, (2) seeking learning opportunities and experiences to understand worldviews during health crisis, (3) enhancing personal growth and psychological development such as through the fulfillment of calling and passion pursuit, (4) gaining career-related clinical skills and experiences, (5) fortifying social relationships with peers and beneficiaries (e.g. migrant workers), and (6) protecting oneself such as from feeling bored, purposeless and guilty for not helping.”

Reviewer 2 Report
Dear authors, Your revisions adequately address the issues, following which the authors should be able to do the final touch on language and style issues.
Author Response
Response: Thank you for the feedback. The revised paper has been proofread.